# Event Causality Extraction via Implicit Cause-Effect Interactions

**Jintao Liu[1,2], Zequn Zhang[1,2]\*, Kaiwen Wei[1,2], Zhi Guo[1,2], Xian Sun[1,2],**
**Li Jin[1,2], Xiaoyu Li[1,2]**

[1]Key Laboratory of Network Information System Technology, Aerospace
Information Research Institute, Chinese Academy of Sciences
[2]School of Electronic, Electrical and Communication Engineering, University of Chinese
Academy of Sciences, Beijing, China
liujintao201@mails.ucas.ac.cn, zqzhang1@mail.ie.ac.cn

## Abstract

Event Causality Extraction (ECE) aims to extract the cause-effect event pairs from the given text, which requires the model to possess a strong reasoning ability to capture event causalities. However, existing works have not adequately exploited the interactions between the cause and effect event that could provide crucial clues for causality reasoning. To this end, we propose an Implicit Cause-Effect interaction (ICE) framework, which formulates ECE as a template-based conditional generation problem. The proposed method captures the implicit intra- and inter-event interactions by incorporating the privileged information (ground truth event types and arguments) for reasoning, and a knowledge distillation mechanism is introduced to alleviate the unavailability of privileged information in the test stage. Furthermore, to facilitate knowledge transfer from teacher to student, we design an event-level alignment strategy named Cause-Effect Optimal Transport (CEOT) to strengthen the semantic interactions of cause-effect event types and arguments. Experimental results indicate that ICE achieves state-of-the-art performance on the ECE-CCKS dataset.

## 1 Introduction

Event Causality Extraction (ECE) is an emerging yet challenging natural language processing (NLP) task, which requires extracting the whole event structure of cause and effect events, including event type and event arguments. As shown in Figure 1, given the input text, an ECE system is expected to identify the cause event (i.e., the event type is *Price Increase*, the argument *oil* plays the role of *Product*, the argument *worldwide* as *Region*), and similarly identify the corresponding effect event. Recognizing event causality can provide support for many downstream NLP tasks, including machine reading comprehension (Berant et al., 2014),

---

\*Corresponding author.

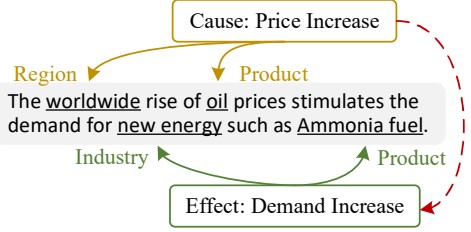

Figure 1: An instance of event causality extraction. The cause event is marked in yellow and the effect event is marked in green. The cause-effect interactions include intra- and inter-event interactions. Solid lines indicate intra-event interactions between the event type and the corresponding arguments, while the dashed line indicates inter-event interaction between the cause-effect event.

| Cause: Price Increase | |
|---|---|
| Region: | worldwide |
| Industry: | None |
| Product: | oil |

| Effect: Demand Increase | |
|---|---|
| Region: | None |
| Industry: | new energy |
| Product: | Ammonia fuel |

intelligent search (Rudnik et al., 2019), future event forecasting (Hashimoto, 2019), and why-question answering (Oh et al., 2017).

Existing studies mainly regard this task as a classification-based problem. Du and Cardie (2020) and Wang et al. (2019) extracted the events and classified the causal relation in a pipelined manner. Wang et al. (2020) leveraged grid tagging to simultaneously extract the events and causality pairs. Cui et al. (2022) also designed a dual grid tagging scheme, which aims at modeling the correlations between different event arguments. Compared to the classification-based methods, generation-based models can take full advantage of Pre-trained Language Models (PLMs) by designing flexible prompt templates (Ma et al., 2022; Liu et al., 2022). As a result, there is a trend to cast the extraction task as a conditional generation problem.

Although generation-based approaches have

achieved remarkable success, the limited interactions between the cause-effect events impede the model's ability to reason effectively between events. Intuitively, the privileged information (Xu et al., 2020), which stands for the ground truth information of event types or event arguments, could provide valuable knowledge for inferring causal clues. Take Figure 1 as an example, if the model has already known the cause event is *Price Increase*, and the *Product* and *Region* are *oil* and *worldwide* in this event, the inter-event interactions could benefit the extraction of effect event. Similarly, if the model has known *Price Increase* causes *Demand Increase* during training, the intra-event interactions may help the model to capture event arguments more accurately. By incorporating different kinds of privileged information, the model could make full use of the implicit interactions and be guided to extract causal clues. However, due to the unavailability of such privileged information in practice, incorporating it naively will result in inconsistencies between training and test phases and may affect the model performance. Several methods (Liu et al., 2017; Wei et al., 2021) in other NLP fields have provided solutions to overcome this problem, but they are not applicable when ECE is modeled as a generative problem.

Moreover, generation-based methods are typically trained via maximum likelihood estimation (MLE) (Salakhutdinov, 2015), which maximizes the likelihood of the next word conditioned on its previous ground truth words. Then, it leverages cross-entropy loss to measure the difference at each position of the target sequence. Nevertheless, since MLE only emphasizes strict word-level alignment, it struggles to consider the semantic information from the perspective of event type or event arguments. For instance, as shown in Figure 1, when training the word *new* from the effect event argument *new energy*, MLE ignores *new energy* is a whole unit as the *Industry*, which results in a partial loss of semantics. The event type and event arguments from the cause-effect event pairs could also be regarded as different wholes, and by interacting with each other, the model could implicitly incorporate such event-level semantic information.

In this paper, we propose an Implicit Cause-Effect interaction (ICE) framework for ECE to address the above issues. Specifically, we formulate the ECE task as a template-based conditional generation problem, which takes the context and prompt template as the input, and decodes event causality and event structure from the generated sequence. To capture the implicit intra- and inter-event interactions, we feed different privileged information to the input template and train two well-informed teacher models. Then a student model is driven by imitating the behaviors of teachers to narrow the input difference of training and test phases through knowledge distillation (Hinton et al., 2015). Furthermore, to facilitate knowledge transfer and strengthen the interactions between cause and effect events, we design a Cause-Effect Optimal Transport (CEOT) mechanism by treating the event type and event arguments as model units, which could implicitly incorporate the event-level semantic information. In summary, the contributions of this paper are as follows:

1) This work proposes an ICE framework, which models event causality extraction in a generative paradigm and incorporates privileged knowledge for reasoning.

2) The proposed method implicitly captures the intra- and inter-event interactions through knowledge distillation, and employs a CEOT strategy to strengthen the semantic interactions of cause and effect events.

3) Experimental results show that our model achieves state-of-the-art performance, improving the F1-score by 8.39% on the ECE evaluation benchmark.

## 2  Related Work

**Event Causality Extraction.** ECE is derived from the previous event causality identification (ECI), which aims to recognize the causal relations between the given events in text (Zuo et al., 2021; Phu and Nguyen, 2021). Early methods mainly focus on syntactic and lexical features (Gao et al., 2019), causality patterns (Hidey and McKeown, 2016), and statistical causal clues (Hu and Walker, 2017). Recent works seek to employ external knowledge (Liu et al., 2020; Cao et al., 2021) or prompt-based models (Shen et al., 2022; Liu et al., 2023) for this task. But these methods only identify the causality of events expressed by a word or phrase, without considering the event type and event arguments. Cui et al. (2022) first proposed the ECE task and exploited the argument correlations to extract event causality and event structure. Some variant methods from relation extraction have also been applied to this task (Wang et al., 2020; Wei et al., 2020).

However, these works fail to model the implicit cause-effect interactions, making it difficult to extract causal clues.

**Knowledge Distillation.** This technique is first proposed by Hinton et al. (2015), which aims to transfer knowledge from a well-trained teacher to a student model. Jiao et al. (2020) and Sanh et al. (2019) used knowledge distillation for compressing large-scale pre-trained language models. Wu et al. (2021) and Li et al. (2022) adopted multiple-teacher knowledge distillation to improve the effectiveness of distillation. Wei et al. (2023a) proposed to incorporate related arguments knowledge through knowledge distillation for event argument extraction. Nevertheless, they are designed for classification-based methods and struggle to migrate to the ECE task under the generative pattern.

**Optimal Transport.** OT has a wide range of applications in NLP domains (Chen et al., 2019; Xu et al., 2021; Wei et al., 2023b). Li et al. (2020) proposed using optimal transport to tackle the exposure bias issue in training generative models by maximum likelihood estimation. Zhou et al. (2022) modeled events in the sequence as units and adopted optimal transport to explicitly extract the event semantics for generating temporally-ordered event sequences. In this paper, we employ optimal transport to improve the semantic interactions of event type and event argument for cause and effect events.

## 3 Methodology

Our ICE framework formulates ECE as a template-based generation problem and implicitly incorporates privileged information for reasoning. Under this paradigm, we train two well-informed teacher models by incorporating different privileged information into model inputs. Then we adopt a knowledge distillation mechanism to drive a student model to capture implicit cause-effect interactions, which could alleviate the difference of unavailable privileged information in the test stage. During the training phase, a CEOT strategy is adopted to improve the semantic interactions of cause-effect events and promote the training of the student. The overview of ICE is shown in Figure 2.

### 3.1 Task Formulation

The goal of ECE is to extract event causality and event structure from the text. Formally, given a context, ECE aims to extract a set of cause-effect event pairs $\{E_{ca_i}, E_{ef_i}\}_{i=1}^s$, where $E_{ca_i}$ and $E_{ef_i}$ indicate the cause and effect event of the $i$-th pair, respectively. The event structure $E = (t, A)$ contains the event type $t$ and the argument set $A$. Each argument in $A$ corresponds to a role.

### 3.2 Generative Template-based ECE Model

**Template Creation.** At the input stage, we first construct a specific task-related template for ECE. Following Ma et al. (2022) and Liu et al. (2021), we design a soft template that contains learnable pseudo tokens and slots for all components we require extracting. Figure 2(c) shows the ECE template in our model, where <Cause>, <Effect>, <type>, </type>, etc. are specific learnable pseudo tokens. Then we concatenate the context and template, and feed them into a Transformer-based model to generate the output sequence, where the slots in the template will be filled with concrete event type or event arguments of the cause event and effect event.

**Target Output Sequence.** For the cause-effect event pair in context, we construct the target output sequence $Y$ for conditional generation by filling the ground truth cause event and effect event into the template. Note that when there is more than one argument corresponding to a role, they will be concatenated by a special token <and>. If some roles of the event have no arguments, the corresponding positions in the target output sequence will be filled with <None>.

**Training.** In the training process, we adopt the Transformer-based pre-trained language model BART (Lewis et al., 2020) as our basic model architecture, which consists an encoder and a decoder.

$$\begin{aligned} \boldsymbol{H}_e &= \mathrm{Encoder}(X) \\ \boldsymbol{H}_d &= \mathrm{Decoder}(Y; \boldsymbol{H}_e) \end{aligned} \tag{1}$$

where $X$ denotes the concatenation of context and template. The training target is to maximize the likelihood of the next token conditioned on the previous ground truth tokens in the sequence:

$$\mathcal{L}_{gen} = -\sum_{l=1}^{L} \log p(Y_l | Y_{<l}, X) \tag{2}$$

where $L$ is the length of the target output sequence.

**Inference.** After obtaining the generated sequence, we decode the event type and event arguments of cause-effect events from corresponding slots with a rule-based matching algorithm. Then we check

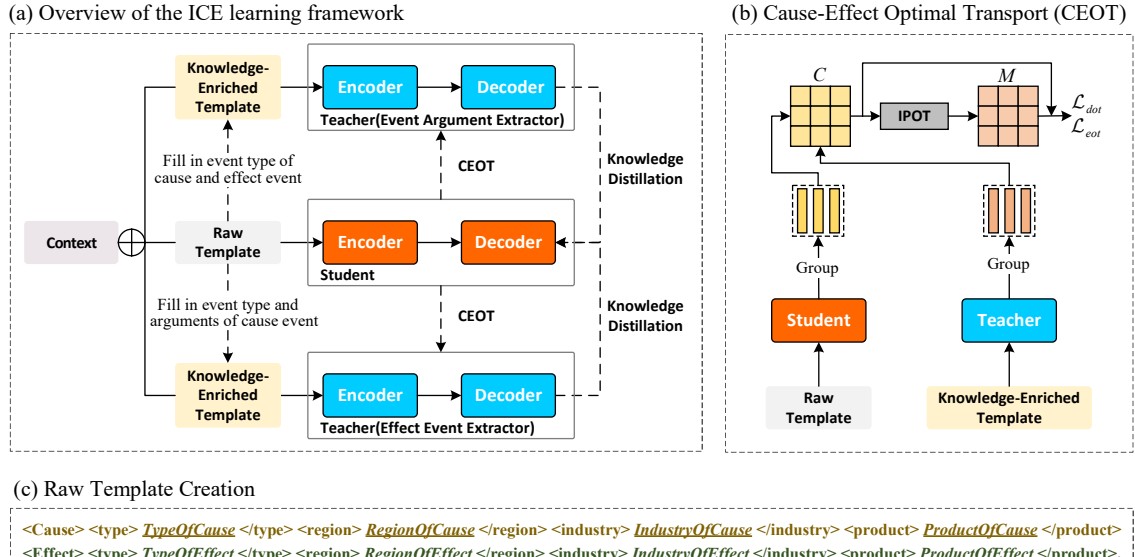

(a) Overview of the ICE learning framework

(b) Cause-Effect Optimal Transport (CEOT)

(c) Raw Template Creation

<Cause> <type> *TypeOfCause* </type> <region> *RegionOfCause* </region> <industry> *IndustryOfCause* </industry> <product> *ProductOfCause* </product>
<Effect> <type> *TypeOfEffect* </type> <region> *RegionOfEffect* </region> <industry> *IndustryOfEffect* </industry> <product> *ProductOfEffect* </product>.

Figure 2: (a) The overview of the ICE learning framework. (b) The process of Cause-Effect Optimal Transport (CEOT). (c) The creation of the template. The cause event is marked in orange and the effect event is marked in green. Underlined tokens indicate slots for event types and event arguments of cause-effect events. And tokens with <angle brackets> denote specific pseudo tokens.

whether the event type is in the predefined event type set and whether each event argument is a span of the context.

### 3.3 Teacher-Student Distillation Learning

Based on the generative paradigm, we adopt a teacher-student distillation learning framework to capture the implicit intra- and inter-event interactions, which could implicitly incorporate the privileged information for event causality reasoning and narrow the inconsistencies between training and test phases.

Specifically, we train two well-informed teachers, including an event argument extractor and an effect event extractor. The event argument extractor aims to extract event arguments with the event type of cause event and effect event given, and the effect event extractor seeks to recognize event type and event arguments of effect event with knowing the information of cause event. With template-based generation formulation, we could introduce the privileged information into the model in a flexible way. As shown in Figure 2, for the event argument extractor, we construct a knowledge-enriched template by filling the ground truth event types of the cause and effect event in the raw template. Likewise, for the effect event extractor, we fill in the ground truth event type and arguments of the cause event to form a new knowledge-enriched template. We give an example for the

construction of knowledge-enriched templates in Appendix B. Next, we concatenate the context and the knowledge-enriched templates as inputs to train teachers.

In the knowledge distillation stage, we use $\boldsymbol{H}_{d_i}^S$ to denote the hidden states of the $i$-th decoder layer of the student model, and $\boldsymbol{H}_{d_i}^T$ to denote the teacher's. We adopt the mean squared error (MSE) to encourage the student model to match the hidden states of corresponding layers of the decoder in the teacher:

$$\mathcal{L}_{mse} = \sum_{l=1}^{L} \sum_{i=1}^{N} \text{MSE}(\boldsymbol{H}_{d_i}^S, \boldsymbol{H}_{d_i}^T) \qquad (3)$$

where $N$ is the number of decoder layers. We also employ KL-Divergence to encourage the student to match the probability distribution of the teacher over the next possible word at each position:

$$\mathcal{L}_{kl} = \sum_{l=1}^{L} \text{KL}(p_l^S, p_l^T) \qquad (4)$$

where $p_l^S$ and $p_l^T$ are probability distributions of student and teacher over the next possible token at position $l$. Please note that the teachers and student exploit the same model architecture and training objectives, but do not share model parameters. And the parameters of teachers are fixed during the training of the student. The overall loss for training

student model with a single teacher is:

$$\mathcal{L}_{kd} = \mathcal{L}_{gen} + \alpha\mathcal{L}_{mse} + \beta\mathcal{L}_{kl} \qquad (5)$$

where $\alpha$ and $\beta$ are weight coefficients.

### 3.4 Cause-Effect Optimal Transport

To improve knowledge transfer, we seek to model semantic information from the perspective of event type or event arguments and introduce a CEOT strategy to promote interactions of cause and effect events, which is achieved by event-level alignment of teacher and student representations.

Optimal transport defines a distance metric between two probability measures on a domain. Given two discrete probability measures $\boldsymbol{\mu} = \sum_{i=1}^{n} u_i \delta_{\boldsymbol{x}_i}$ and $\boldsymbol{\nu} = \sum_{j=1}^{m} v_j \delta_{\boldsymbol{y}_j}$, where $\delta_{\boldsymbol{x}_i}$ is the Dirac function centered on $\boldsymbol{x}$, the weights $\boldsymbol{u} = \{u_i\}_{i=1}^{n} \in \Delta_n$ and $\boldsymbol{v} = \{v_j\}_{j=1}^{m} \in \Delta_m$ satisfy the constraints $\sum_{i=1}^{n} u_i = \sum_{j=1}^{m} v_j = 1$. Under this setting, the OT distance is formalized as the following problem (Luise et al., 2018):

$$\begin{aligned} \mathcal{L}_{ot}(\boldsymbol{\mu}, \boldsymbol{\nu}) &= \min_{\boldsymbol{M} \in \Pi(\boldsymbol{u}, \boldsymbol{v})} \sum_{i=1}^{n} \sum_{j=1}^{m} \boldsymbol{M}_{ij} \cdot c(\boldsymbol{x}_i, \boldsymbol{y}_j) \\ &= \min_{\boldsymbol{M} \in \Pi(\boldsymbol{u}, \boldsymbol{v})} \langle \boldsymbol{M}, \boldsymbol{C} \rangle \end{aligned}$$
$$(6)$$

where $\Pi(\boldsymbol{u}, \boldsymbol{v}) = \{\boldsymbol{M} \in \mathbb{R}_+^{n \times m} | \boldsymbol{M}\mathbf{1}_m = \boldsymbol{u}, \boldsymbol{M}^\top \mathbf{1}_n = \boldsymbol{v}\}$, $\mathbf{1}_n$ denotes an $n$-dimensional all-one vector, $\boldsymbol{C}$ is cost matrix defined as $\boldsymbol{C}_{ij} = c(\boldsymbol{x}_i, \boldsymbol{y}_j)$, $\boldsymbol{M}$ is the transportation plan, and $\langle \boldsymbol{M}, \boldsymbol{C} \rangle = \text{Tr}(\boldsymbol{M}^\top \boldsymbol{C})$ denotes the Frobenius inner product. Xie et al. (2019) proposed an approximate algorithm IPOT to solve Eq. 6, which is illustrated in Appendix A. After solving $\boldsymbol{M}$, we use OT distance as loss to update model parameters.

Specifically, the representation of the template for student is denoted as $\boldsymbol{H}_e^S$, which is obtained from the last hidden state of the BART encoder of student corresponding to the template. We first partition $\boldsymbol{H}_e^S$ into $n$ groups, where each group corresponds to the representation of the slot together with specific tokens before and behind it (e.g., <type> *TypeOfCause* </type> belong to a group). Then we average the representations of tokens in each group to obtain the sequence $\boldsymbol{K}_e^S = \{\boldsymbol{h}_{e_i}^S\}_{i=1}^n$, where $\boldsymbol{h}_{e_i}$ denotes the representation of the $i$-th group, $n$ is the number of groups. Similarly, we can get the sequence $\boldsymbol{K}_e^T = \{\boldsymbol{h}_{e_i}^T\}_{i=1}^m$ by group and average the template representations of teacher.

| Split | #Sents | #Pairs | #Events |
|-------|--------|--------|---------|
| Train | 5600 | 6318 | 12636 |
| Dev | 700 | 791 | 1582 |
| Test | 700 | 799 | 1598 |

Table 1: Statistics of the ECE-CCKS dataset.

We use cosine distance as the cost function and adopt the IPOT algorithm to compute the OT loss:

$$\mathcal{L}_{eot} = \text{IPOT}(\boldsymbol{K}_e^S, \boldsymbol{K}_e^T) \qquad (7)$$

Meanwhile, the last hidden state of the BART decoder for teacher and student is denoted as $\boldsymbol{H}_d^T$ and $\boldsymbol{H}_d^S$. We first use a linear layer followed by an *argmax* function to decode the output sequence. Then the representations are divided into several groups based on the event type or event argument together with specific tokens before and behind it. Note that when some specific tokens are missing in the output, we remove the corresponding groups. Likewise, we can obtain two sequences $\boldsymbol{K}_d^S$ and $\boldsymbol{K}_d^T$ via an average operation and compute the OT loss:

$$\mathcal{L}_{dot} = \text{IPOT}(\boldsymbol{K}_d^S, \boldsymbol{K}_d^T) \qquad (8)$$

The overall loss of the model is defined as:

$$\mathcal{L}_o = \mathcal{L}_{kd} + \lambda\mathcal{L}_{eot} + \eta\mathcal{L}_{dot} \qquad (9)$$

where $\lambda$ and $\eta$ are weight coefficients.

### 3.5 Training Objectives

Since the two teachers exhibit different difficulties in extracting the same sample, motivated by Zhang et al. (2022), we use adaptive weights to control the importance of different teachers for a specific sample:

$$w_k = 1 - \frac{\exp(\mathcal{L}_{gen_T}^k)}{\sum_j \exp(\mathcal{L}_{gen_T}^j)} \qquad (10)$$

where $\mathcal{L}_{gen_T}^k$ denotes the prediction loss of the $k$-th teacher. The total loss under the multi-teacher knowledge distillation framework is calculated as:

$$\mathcal{L} = \sum_k w_k \mathcal{L}_o^k \qquad (11)$$

## 4 Experiments

### 4.1 Experiment Setup

**Dataset.** Our experiments are conducted on the ECE-CCKS (Cui et al., 2022) dataset, which is

| Method | EAE | | | CET | | | ECE | | |
|---|---|---|---|---|---|---|---|---|---|
| | P | R | F1 | P | R | F1 | P | R | F1 |
| Novel-tagging | 59.40 | 28.47 | 38.49 | 49.79 | 61.70 | 55.11 | 51.52 | 26.75 | 35.22 |
| CasECE | 36.88 | 36.72 | 36.80 | 58.26 | 59.70 | 58.97 | 31.30 | 41.81 | 35.80 |
| Pair-tagging | 47.08 | 46.49 | 46.79 | 55.78 | **62.95** | 59.14 | 39.24 | 47.69 | 43.05 |
| DualCor | 58.05 | 47.60 | 52.31 | 61.75 | 58.19 | 59.92 | 48.56 | 44.85 | 46.63 |
| BART-ECE | 64.62 | 50.09 | 56.43 | 69.43 | 60.83 | 64.84 | 54.74 | 47.21 | 50.70 |
| Student | 67.81 | 54.37 | 60.35 | 67.57 | 59.20 | 63.11 | 55.47 | 49.76 | 52.46 |
| MKD | 68.83 | 53.59 | 60.26 | 68.29 | 59.82 | 63.78 | 57.02 | 49.04 | 52.73 |
| CE-Pipeline | 66.50 | 52.69 | 58.80 | 69.00 | 60.45 | 64.44 | 55.01 | 48.36 | 51.47 |
| TA-Pipeline | 67.01 | 53.55 | 59.53 | 68.86 | 60.33 | 64.31 | 54.27 | 47.98 | 50.93 |
| ICE (Ours) | **69.69** | **54.57** | **61.21** | **70.00** | 61.33 | **65.38** | **59.13** | **51.44** | **55.02** |

Table 2: Overall performance compared to the state-of-the-art methods on the test set. P, R, and F1 denote precision (%), recall (%), and F1-score (%). The best results are denoted in bold.

derived from the corpus released by Tianchi (2021). The dataset is annotated with 39 event types and 3 event roles, and the statistic information is listed in Table 1.

**Evaluation Metric.** We use precision (P), recall (R), and F1-score (F1) as evaluation metrics. A predicted cause-effect event pair is considered correct when the event type and event arguments of the cause event and effect event are correctly extracted. To prove a fair comparison with previous methods (Cui et al., 2022), we also report results on the following two tasks: **Event Argument Extraction (EAE)**, which measures the model's ability to extract event arguments; **Cause-Effect Type extraction (CET)**, which aims to recognize whether the predicted event types of cause and effect event are correct.

**Implementation Details.** All experiments are conducted on NVIDIA Tesla V100 GPU with Pytorch framework. We use the pre-trained *BART-base* from Hugging-Face's Transformers library (Wolf et al., 2020) as the encoder-decoder language model. Our model is optimized by the AdamW weight decay strategy with a learning rate of 3e-5. The coefficient $\lambda$ is set to 0.1 and $\eta$ is set to 0.1. We set $\alpha$ and $\beta$ to 1e-3 and 0.5. The model is trained for 60 epochs with a batch size of 16.

### 4.2 Baseline Methods

**Classification-based Method.** (1) **Novel-tagging** is introduced to ECE by combining causality, event types, event roles, and argument span into a unified label space (Zheng et al., 2017). (2) **CasECE** is a pipelined method inspired by Wei et al. (2020), which first extracts the cause event and then recognizes the effect event conditioned on the former prediction. (3) **Pair-linking** is a grid tagging method based on Wang et al. (2020), which uses event-type-level pair linking as conditional information for token-pair linking to extract event arguments. (4) **DualCor** (Cui et al., 2022) designed a dual grid tagging scheme to capture the argument correlations for ECE.

**Generation-based Method.** Since there is no generative method for ECE in existing studies, we implement the following baselines: (1) **BART-ECE** achieves event causality extraction with a template-based conditional generation method in natural language form. (2) **Student** is the generative model introduced in Section 3.2. (3) **MKD** employs a multi-teacher knowledge distillation framework where each teacher is trained without privileged information. (4) **CE-Pipeline** is a pipelined generation model, which first extracts the cause event and then predicts the effect event based on the cause. (5) **TA-Pipeline** is also a pipelined generation model that first identifies the event type of the cause and effect event, conditioned on which to detect the event arguments.

### 4.3 Overall Performance

The experimental results are reported in Table 2. We can draw the following conclusions: (1) The proposed method achieves the best performance, outperforming the previous state-of-the-art model, DualCor, by 8.90%, 5.46%, and 8.39% on EAE, CET, and ECE in terms of F1-score. The significant improvements demonstrate the effectiveness of our

| Method | EAE | CET | ECE |
|---|---|---|---|
| ICE | **61.21** | **65.38** | **55.02** |
| -w/o EAE | 60.43 | 65.11 | 53.20 |
| -w/o EEE | 60.97 | 64.71 | 53.51 |
| -w/o EAE&EEE | 60.35 | 63.11 | 52.46 |
| -w/o CEOT | 60.97 | 65.18 | 54.06 |
| -w/o AW | 60.45 | 65.11 | 54.30 |

Table 3: Experimental results (F1-score) of ablation study on the test set.

| Pairs | Method | P | R | F1 |
|---|---|---|---|---|
| Single | Pair-linking | 40.31 | 54.32 | 46.28 |
| | DualCor | 49.39 | 51.56 | 50.46 |
| | Student | 55.83 | 55.58 | 55.71 |
| | ICE | **57.74** | **57.78** | **57.76** |
| Multiple | Pair-linking | 32.15 | 24.82 | 28.03 |
| | DualCor | 43.65 | 22.72 | 29.89 |
| | Student | 59.91 | 27.48 | 37.68 |
| | ICE | **63.06** | **28.93** | **39.66** |

Table 4: Experimental results under different number of cause-effect event pairs in a sample on ECE task.

| Temp. | EAE | CET | ECE |
|---|---|---|---|
| CA | **61.88** | 64.84 | 53.56 |
| MA | 60.95 | 64.44 | 54.27 |
| SF | 61.21 | **65.38** | **55.02** |

Table 5: Experimental results (F1-score) of using different types of templates. **CA**: Concatenation Template, **MA**: Manual Template, **SF**: Soft Template.

model. (2) The generation-based methods generally produce better performance than classification-based methods. This suggests that the generation-based model could make full use of the knowledge in PLMs and exhibit strong advantages on ECE. (3) Compared with Student, our method performs better on the three tasks. We credit the reason to that the student model has difficulty in extracting implicit causal clues, while ICE can equip the model with more implicit event causality reasoning knowledge via a multi-teacher knowledge distillation framework, thus boosting the model performance. (4) With the same model architecture, the performance of ICE exceeds MKD. The improvements indicate that the privileged information could help to train well-informed teachers, which guide the student to capture the intra- and inter-event interactions.(5) CE-Pipeline and TA-Pipeline perform poorly among the generation-based methods. The results illustrate that the pipelined methods suffer from error accumulation problems, while ICE is an end-to-end method with privileged information as the supervision, which could avoid introducing noise or irrelevant information.

### 4.4 Further Discussion

**Ablation Study.** To evaluate the contribution of each component, we conduct ablation studies by removing event argument extractor (w/o EAE), effect event extractor (w/o EEE), Cause-Effect Optimal Transport (w/o CEOT), and adaptive weights (w/o WA). The experimental results are shown in Table 3. We observe that: (1) Removing the event argument extractor or effect event extractor will result in performance decay, demonstrating that both extractors are beneficial for event causality extraction. This is because they could capture the implicit intra- and inter-event interactions to perform event causality reasoning, and the knowledge is transferred to the student via a teacher-student

distillation framework. (2) The model performance decreases after removing CEOT, which illustrates this strategy could promote the training process by enhancing the event-level semantic information interactions of cause event and effect event. (3) The adaptive weights mechanism can assign different reliability for each teacher according to the characteristics of the samples, further improving the performance of the student.

**Effect of the Number of Cause-Effect Pairs.** To evaluate the effect of different numbers of event causality pairs in a sample, we conduct experiments by dividing the test set into two subsets: **Single**, which indicates the subset with one pair in a sample; **Multiple**, which denotes the subset with more than one pairs. The results are reported in Table 4. It can be observed that: (1) ICE achieves the best performance among all the baselines on Single and Multiple subsets, which shows the generalization ability and robustness of ICE. (2) The methods on Multiple generally suffer from weak performance. The reason may be that multiple event causality pairs impose difficulties in capturing implicit causal clues, leading to extremely low recall. (3) The performance gaps between Student and ICE on the two subsets further demonstrate that the knowledge distillation framework could utilize the privileged information to improve the

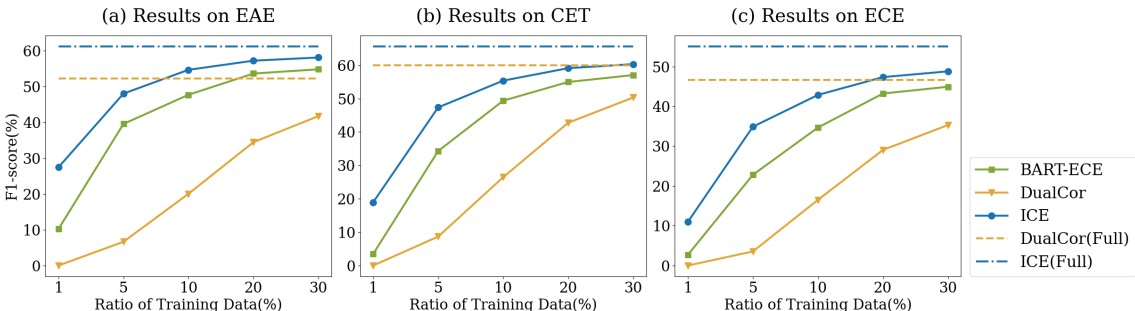

Figure 3: Experimental results (F1-score) under different ratios of training data on three tasks.

| | |
|---|---|
| **Example1** | The growth of {global}*RegionOfCause* {gasoline}*ProductOfCause* demand is sluggish, and the combination of excess supply and unsatisfactory demand has led to a continuous decline in {gasoline}*ProductOfEffect* cracking spreads this year.
**BART-ECE**: {*TypeOfCause*: Supply Increase, *TypeOfEffect*: Demand Drop}
**ICE**: {*TypeOfCause*: Demand Drop, *TypeOfEffect*: Price Drop} |
| **Example2** | The amount of {silicon}*ProductOfCause* used in June increased compared to May, and the increase came from polysilicon, while the supply of {monosilicon}*ProductOfCause* decreased. Therefore, it is expected that the price of {monosilicon}*ProductOfEffect* will rise after July.
**BART-ECE**: {*TypeOfCause*: Supply Decrease, *TypeOfEffect*: Price Increase}
**ICE**: {*TypeOfCause*: Supply Decrease, *TypeOfEffect*: Price Increase} |

Table 6: Case study on the test set. The correctly predicted event type or event arguments are marked in teal, while wrong predictions by BART-ECE are marked in red.

event causality reasoning ability of the model, thus driving a more effective model.

**Performance in Low-resource Scenarios.** To investigate the model performance in low-resource scenarios, we adopt different low proportions of training data to conduct experiments on the three tasks. As shown in Figure 3, it can be observed that: (1) ICE achieves the best performance under different ratios of training data on three tasks. This suggests that our model can extract event causalities and event structure effectively with a small scale of annotated data, which is more practical to use. (2) ICE using a small amount of data can even exceed DualCor using full data, and the performance gap becomes larger with the decrease of training data. These observations indicate that ICE can elicit knowledge in PLMs and has a strong ability to capture causal clues, which is beneficial for the model to perform the ECE task.

**Effect of the Prompt Template.** We study how different types of prompt templates affect the model performance by conducting experiments with the following templates: Concatenation Template, Manual Template, and Soft Template. The creation of the three templates is shown in Appendix C and the results are listed in Table 5. We find that the soft template is superior to the manual template and concatenation template, which illustrates the effectiveness of our template. Although the manual template can elicit pre-training knowledge in a cloze formulation, it is labor-intensive and hard to achieve optimal. However, the soft template can avoid this laborious process and take the best advantage of the PLMs.

**Case Study.** We present case studies to further illustrate the performance of the proposed method. As shown in Table 6, for Example1, BART-GEN gives wrong predictions about the event type of the cause event and effect event. The reason may be that BART-ECE has difficulty capturing implicit cause-effect interactions, so it fails to recognize the causality between *Demand Drop* and *Price Drop*. For Example2, both methods produce the correct cause-effect event type, while BART-GEN fails to predict the event argument *monosilicon* of the cause event. We credit the reason to that our model could leverage the event argument extractor and effect event extractor trained with privileged information to guide the training process of the student, thus obtaining better performance in extracting event causality and event structure.

## 5 Conclusions

In this paper, we propose an Implicit Cause-Effect interaction (ICE) framework to improve the reasoning ability of the model, which tackles ECE in a generative manner. The proposed method incorporates privileged information for reasoning to capture implicit intra- and inter-event interactions, and utilizes a teacher-student learning framework to bridge the gap between training and test stages. Besides, we introduce a Cause-Effect Optimal Transport (CEOT) strategy to improve the event-level semantic interactions of cause and effect events. Experimental results indicate that ICE outperforms all the baselines on the ECE-CCKS dataset, demonstrating the effectiveness of this work.

## Limitations

The multi-teacher knowledge distillation mechanism utilized in the ICE framework may increase the computational time during the training process. However, only the student model is leveraged during the test process, and the test time is identical to regular generation-based models. The problem of relatively long training time can be mitigated by strategies such as GPU parallelization. Considering the significant improvement brought by the ICE framework, we believe the cost is acceptable.

## Acknowledgements

The research is supported by the National Natural Science Foundation of China under Grant 62206267.

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

## A The details of the IPOT algorithm

The details of the IPOT algorithm is shown in Algorithm 1.

---

**Algorithm 1** IPOT algorithm

---

**Input:** Feature vectors $S = \{h_i\}_1^n$, $S' = \{h_j'\}_1^m$, Generalized stepsize $\frac{1}{\beta}$

**Output:** $\langle M, C \rangle$

1: $\sigma = \frac{1}{m}\mathbf{1}_m$, $M^{(1)} = \mathbf{1}_n\mathbf{1}_m^\top$
2: $C_{ij} = c(h_i, h_j')$, $A_{ij} = e^{-\frac{C_{ij}}{\beta}}$
3: **for** $t = 1, 2, 3 \cdots$ **do**
4:    $Q = A \odot M^{(t)}$ // $\odot$ is Hadamart product
5:    **for** $k = 1, 2, \cdots, K$ **do** //$K$=1 in practice
6:       $\delta = \frac{1}{nQ\sigma}, \sigma = \frac{1}{mQ^\top\delta}$
7:    **end for**
8:    $M^{(t+1)} = \text{diag}(\delta)\, Q\, \text{diag}(\sigma)$
9: **end for**
10: **return** $\langle M, C \rangle$

---

## B An example of the construction of knowledge-enriched templates

We show an example of the construction of knowledge-enriched templates for EAE and EEE in Table 7. The knowledge-enriched template for EAE is constructed by filling the ground truth event types of the cause and effect event in the raw template; The knowledge-enriched template for EEE is constructed by filling the ground truth event type and arguments of the cause event in the raw template.

## C The creation of different types of templates

We construct three types of templates for comparison: (1) Concatenation Template, where all slots for event type and event arguments are concatenated; (2) Manual Template, where event types and event arguments are integrated into templates in natural language form; (3) Soft Template, which is used in our method. And detailed information of the templates is shown in Table 8.

## D Compare with ChatGPT

In this section, we conduct experiments to evaluate the performance of ChatGPT on the ECE task. A well-designed prompt template is as follows:

Suppose you are now an event causality extraction model. Given a sentence, please give the cause event and result event respectively, where the event contains the event type and the arguments corresponding to each role. The list of event types is: ['Typhoon', 'Demand Increase', 'Price Decrease', 'Cold Wave', 'Price Increase', 'Other Natural Disasters', 'Supply Decrease', 'Supply Increase', 'Sales Decrease', 'Demand Drop', 'Import Decrease', 'Flood', 'Other Trade Frictions', 'Negative Impact', 'Swine Fever', 'Sales Increase', 'Limited Production', 'Operating Costs Increased', 'Other Livestock Epidemics', 'Positive Impact', 'Drought', 'Operating Cost Decrease', 'Export Decrease', 'Frost', 'Other or Unclear', 'Import Increase', 'Bird Flu', 'Earthquake ', 'Anti-dumping Against China', 'Exports Increase', 'Add Tariffs to China', 'Decrease in Product Profits', 'Increase in Product Profits', 'Foot-and-mouth Disease of Pigs', 'Anti-dumping Against Other Countries', 'Unsalable', 'Cattle Foot and Mouth Disease', 'Flash Flood', 'Hail']. The list of event argument roles is: ['Region', 'Industry', 'Product']. Given a sentence:"The worldwide rise of oil prices stimulates the demand for new energy such as Ammonia fuel.", please extract the event type and arguments corresponding to cause and effect event. If no argument corresponds to a role, the argument content returns "None". If multiple arguments corresponds to a role, the arguments are connected with "and".

As shown in Table 9, we can observe that ICE outperforms ChatGPT by a large margin on EAE, CET, and ECE tasks. The results indicate that ChatGPT has difficulty solving such complex event causality extraction tasks without any fine-tuning or training to update parameters.

| Temp. | Example |
|---|---|
| Raw Temp. | <Cause> <type> *TypeOfCause* </type> <region> *RegionOfCause* </region> <industry> *IndustryOfCause* </industry> <product> *ProductOfCause* </product> <Effect> <type> *TypeOfEffect* </type> <region> *RegionOfEffect* </region> <industry> *IndustryOfEffect* </industry> <product> *ProductOfEffect* </product>. |
| Temp. for EAE | <Cause> <type> Price Increase </type> <region> *RegionOfCause* </region> <industry> *IndustryOfCause* </industry> <product> *ProductOfCause* </product> <Effect> <type> Demand Increase </type> <region> *RegionOfEffect* </region> <industry> *IndustryOfEffect* </industry> <product> *ProductOfEffect* </product>. |
| Temp. for EEE | <Cause> <type> Price Increase </type> <region> None </region> <industry> new energy </industry> <product> Ammonia fuel </product> <Effect> <type> *TypeOfEffect* </type> <region> *RegionOfEffect* </region> <industry> *IndustryOfEffect* </industry> <product> *ProductOfEffect* </product>. |

Table 7: An example of the construction of knowledge-enriched templates for EAE and EEE.

| Temp. | Example |
|---|---|
| CA | *TypeOfCause RegionOfCause IndustryOfCause ProductOfCause* *TypeOfEffect RegionOfEffectIndustryOfEffect ProductOfEffect*. |
| MA | The cause *TypeOfCause*, the region *RegionOfCause*, the industry *IndustryOfCause*, the product *ProductOfCause*, leads to the effect *TypeOfEffect*, the region *RegionOfEffect*, the industry *IndustryOfEffect*, the product *ProductOfEffect*. |
| SF | <Cause> <type> *TypeOfCause* </type> <region> *RegionOfCause* </region> <industry> *IndustryOfCause* </industry> <product> *ProductOfCause* </product> <Effect> <type> *TypeOfEffect* </type> <region> *RegionOfEffect* </region> <industry> *IndustryOfEffect* </industry> <product> *ProductOfEffect* </product>. |

Table 8: The creation of different types of templates. **CA**: Concatenation Template, **MA**: Manual Template, **SF**: Soft Template.

| Method | EAE | | | CET | | | ECE | | |
|---|---|---|---|---|---|---|---|---|---|
| | **P** | **R** | **F1** | **P** | **R** | **F1** | **P** | **R** | **F1** |
| ChatGPT | 13.16 | 16.26 | 14.54 | 17.85 | 15.64 | 16.68 | 6.00 | 8.04 | 6.87 |
| ICE | **69.69** | **54.57** | **61.21** | **70.00** | **61.33** | **65.38** | **59.13** | **51.44** | **55.02** |

Table 9: Overall performance compared to ChatGPT on the test set. P, R, and F1 denote precision (%), recall (%), and F1-score (%). The best results are denoted in bold.