# OpenReview forum: "Event Causality Extraction via Implicit Cause-Effect Interactions"
_EMNLP/2023/Conference — EMNLP 2023 Main_

### Official Review · Reviewer_ttgs · 2023-08-04

**Typos Grammar Style And Presentation Improvements:** None
**Soundness:** 4

**Excitement:**

3: Ambivalent: It has merits (e.g., it reports state-of-the-art results, the idea is nice), but there are key weaknesses (e.g., it describes incremental work), and it can significantly benefit from another round of revision. However, I won't object to accepting it if my co-reviewers champion it.

**Missing References:**

None

**Paper Topic And Main Contributions:**

This paper studies the task of Event Causality Extraction(ECE). Following the recent trend, ECE is formulated in a generative paradigm, and the authors aims to improve the model performances from two aspects: 1)  enhancing the limited interaction between cause-effect events; 2) exploring the semantic information of event types or arguments. To this end, two strategies are conducted in this paper: 1) knowledge distillation works to distill the privileged information from the teacher models with well-informed templates to the student model with a raw template. 2) Optimal Transport further facilitates the knowledge transfer between teacher models and student models.

**Questions For The Authors:**

Please refer to the Reasons To Reject.

**Reasons To Accept:**

1.The proposed method is reasonable to tackle the claimed challenge.

2.Figure 2 goes clearly to illustrate the whole model architecture.

3.Experiments on the public benchmark demonstrate the effectiveness of the proposed method.

**Reasons To Reject:**

1.The word “Implicit” in the title is not significantly explained in the paper. I try to explore the clear definition to “Implicit”, but still confused. Is the word ‘”limitted” in Line60 equivalent to “Implicit”？ Besides, Line63-78 just describes how the privileged information/cause-effect interaction helps to guide task predictions, but does not state clearly why such interaction is unavailable or implicit.

2.In Line294 and 299, why is the distillation  conducted with MSE and KL simultaneously? Please add some explaination.

3.Line429-435 explains the BART-ECE and Student implicitly. Is (2) Student the student model with a raw template in Section 3? What is the difference between BART-ECE and the teacher model?

4.Going on with 3, please show the original model performances of the two teacher models and the student model.

**Reproducibility:**

4: Could mostly reproduce the results, but there may be some variation because of sample variance or minor variations in their interpretation of the protocol or method.

**Reviewer Confidence:**

4: Quite sure. I tried to check the important points carefully. It's unlikely, though conceivable, that I missed something that should affect my ratings.

---

> ### Author Rebuttal · Authors · 2023-08-26
>
> Thank you very much. We feel sorry for not explaining some details clearly.
>
> **Q**: Is “Implicit” equivalent to “limited” in the paper？\
> **A**: "Implicit" is the opposite of “explicit”, meaning that there is no obvious marker word to suggest the occurrence of event causality or event arguments, while “limited” refers to the lack of interactions. And event causality and arguments are implicit in most cases.
>
> **Q**: Why privileged information is unavailable?\
> **A**: Since the privileged information represents ground-truth label, it is not available in the testing phase or in real applications. In our method, only the teacher model can use privileged information and guide the training of the student model through knowledge distillation, which enables the student model to effectively perform causality reasoning without privileged information in the testing phase.
>
> **Q**: Why is the distillation conducted with MSE and KL simultaneously?\
> **A**: MSE is used to distill the feature representation of each layer of the decoder, while KL is utilized to distill the output probability. Using these two methods at the same time can not only promote the student model to imitate the representation of each layer of the teacher, but also make the prediction probability of the student model approach the teacher model, thereby promoting the training process of the student model.
>
> **Q**: What is the difference between BART-ECE and the student model?\
> **A**: We adopt the generative model with raw template in Section 3.2 as Student. And BART-ECE uses manual template in natural language form without special pseudo tokens. We will revise the descriptions and add more details.
>
> **Q**: Show the model performances of the two teacher models and the student model.\
> **A**: We report the experimental results in the following table:
> | | P(EAE) | R(EAE) | F1(EAE)| P(CET) |R(CET)| F1(CET)|P(ECE)|R(ECE)|F1(ECE)|
> |-|-|-|-|-|-|-|-|-|-|
> |EAE(teacher) | 79.71|	 61.27|  69.28|  100.00|  87.61|   93.40|  79.21|   68.77 |  73.62|
> |EEE(teacher)  |84.97|  65.54|   73.97|   86.14|  74.47|   80.45|  77.53|   67.56|   72.20|
> |Student| 67.81| 54.37| 60.35| 67.57| 59.20| 63.11| 55.47| 49.76| 52.46|
> |ICE |   69.69| 54.57| 61.21| 70.00| 61.33| 65.38| 59.13| 51.44| 55.02|
>
> EAE(teacher) and EEE(teacher) indicates event argument extractor teacher and effect event extractor teacher, respectively.

---

### Official Review · Reviewer_dU5i · 2023-08-04

**Soundness:** 3

**Excitement:**

4: Strong: This paper deepens the understanding of some phenomenon or lowers the barriers to an existing research direction.

**Paper Topic And Main Contributions:**

This paper proposes an ICE framework that models implicit cause-effect interactions using privileged information and teacher-student learning. A Cause-Effect Optimal Transport technique is introduced to align event representations and strengthen semantic interactions. Extensive experiments demonstrate the performance advantage.

**Questions For The Authors:**

Could an LLM performing SFT, instructing tunning on the ECE dataset, be a strong baseline?

**Reasons To Accept:**

Pos:

* The problem is well motivated - event causality extraction is an important but challenging NLP task, and modeling implicit interactions between events can provide useful reasoning knowledge.

* The proposed ICE framework is novel, incorporating privileged information and teacher-student distillation to capture intra- and inter-event interactions. The Cause-Effect Optimal Transport strategy is also an interesting technique for aligning event representations.

* The experiments are comprehensive, comparing multiple strong baselines on the ECE-CCKS dataset. The results demonstrate clear improvements, especially in low-resource scenarios. The ablation studies also verify the contribution of each model component.

* The writing is clear and easy to follow. The methodology is well-explained.

**Reasons To Reject:**

Neg:

* The effect of different hyper-parameters is under-explored.

* Ablation results show that the incomplete versions are comparable to the full version. Does the performance gain come from a careful tuning BART?

* The results should reflect the variance of the scores from multiple experiments with different seeds.

**Reproducibility:**

3: Could reproduce the results with some difficulty. The settings of parameters are underspecified or subjectively determined; the training/evaluation data are not widely available.

**Reviewer Confidence:**

4: Quite sure. I tried to check the important points carefully. It's unlikely, though conceivable, that I missed something that should affect my ratings.

---

> ### Author Rebuttal · Authors · 2023-08-26
>
> Thank you very much. We feel sorry for the lack of some analysis.
>
> **Q**: The effect of different hyper-parameters is under-explored.\
> **A**: The main hyper-parameters include $\lambda$, $\eta$, $\alpha$, and $\beta$. We conduct experiments to illustrate the effect of each hyper-parameter. \
> We set $\eta$ equal to $\lambda$ and their value is selected from [0.05, 0.1, 0.2].
> |$\eta$&$\lambda$| EAE|CET|ECE|
> |-|-|-|-|
> |0.05| 61.67| 64.58| 54.09|
> |0.1  |61.21| 65.38| 55.02|
> |0.2 |60.82 | 65.51| 54.14|
>
> The value of $\alpha$ is selected from [5e-4, 1e-3, 2e-3].
> |$\alpha$| EAE|CET|ECE|
> |-|-|-|-|
> |5e-4|   61.40| 65.64| 54.78|
> |1e-3 |61.21| 65.38 |55.02|
> |2e-3 |61.48 |65.17 |54.51|
>
> The value of $\beta$ is selected from [0.3, 0.5, 0.7].
> |$\beta$| EAE|CET|ECE|
> |-|-|-|-|
> |0.3| 61.36| 65.38| 54.42|
> |0.5 |61.21 |65.38 |55.02|
> |0.7 |61.46 | 66.44 |54.69|
>
> We will add more experiments to prove the effectiveness of the hyper-parameter selection.
>
> **Q**: Ablation results show that the incomplete versions are comparable to the full version. Does the performance gain come from a careful tuning BART?\
> **A**: All ablation experiments were performed with the same parameters. The variant models in ablation study only remove the corresponding component.
>
> **Q**: The results should reflect the variance of the scores from multiple experiments with different seeds.\
> **A**: We report the average results over 3 runs with different seeds on the main experiments:
> | | P(EAE) | R(EAE) | F1(EAE)| P(CET) |R(CET)| F1(CET)|P(ECE)|R(ECE)|F1(ECE)|
> |-|-|-|-|-|-|-|-|-|-|
> |ICE|69.97$\pm$0.49|52.83$\pm$0.19|61.48$\pm$0.28|70.76$\pm$0.55|61.99$\pm$0.48|66.09$\pm$0.51|59.07$\pm$0.30|51.52$\pm$0.06|55.04$\pm$0.12|
>
>
> **Q**: Could an LLM be a strong baseline?\
> **A**: We have shown the results of ECE with ChatGPT in the following table. The experiments illustrate that ChatGPT fails to effectively perform causality reasoning and accurate extraction of spans. Due to computational resource constraints, it is difficult for us to perform instruction tuning on the ECE task with LLMs. We argue that the proposed method is superior to LLM in terms of time and space complexity. Therefore, LLM fails to serve as a strong baseline for the ECE task.
> | | P(EAE) | R(EAE) | F1(EAE)| P(CET) |R(CET)| F1(CET)|P(ECE)|R(ECE)|F1(ECE)|
> |-|-|-|-|-|-|-|-|-|-|
> |ChatGPT | 13.16 |16.26| 14.54|17.85| 15.64 |16.68| 6.00| 8.04| 6.87|
> |ICE |   69.69| 54.57| 61.21| 70.00| 61.33| 65.38| 59.13| 51.44| 55.02|

---

### Official Review · Reviewer_BTRu · 2023-08-05

**Soundness:** 4

**Excitement:**

4: Strong: This paper deepens the understanding of some phenomenon or lowers the barriers to an existing research direction.

**Paper Topic And Main Contributions:**

This paper proposes a generative method to extract causal relations (cause-effect pairs) from text. Unlike previous methods that rely on extracting causal relation candidates and classifying them, the proposed method directly extract causal relations from text.

It also uses (1) a system of templates which are used to guide the extraction process, and (2) a distillation process in which a student model is trained to mimic the performance of two teacher models which are given additional information during training. Experimental results a provided which show a strong performance gain obtained by the proposed method over state of the art methods.

**Questions For The Authors:**

Q1: How do you obtain the templates used by your proposed method ? Am I right in thinking that they are used by the student model too ? If those aren't obtained automatically, how do you justify your extractive method having access to high-quality information (these templates) while the other baseline methods do not ?

Q2: is the "context" the same as the "text" from which causal relations are extracted ? (l207-208)

**Reasons To Accept:**

The task dealt with is a relatively important task in the NLP field. Alongside that, the proposed model shows strong performance over existing methods.

The proposed method is original and, even if not directly usable in other fields, could be used as a model for other tasks.

Besides one glaring problem (see reasons to reject), the paper is overall well written, and the experimental setup is strong.

**Reasons To Reject:**

The proposed method relies on templates which are used to guide the extraction process. As far as I understood, the templates are provided as input to the student model, i.e. the one used to evaluate the method. The authors do not explain at all how these templates, which are a crucial component of the proposed method, are obtained.

The paper severely lacks a detailed overall description of the proposed method (the explanation at lines 191-207 is quite short), and most of all actual examples of what is given as input to which model, what is obtained as output, how the output is parsed... As it stands, the proposed method is hard to understand and some details (like details of the rule-matching algorithm) are missing.

**Reproducibility:**

3: Could reproduce the results with some difficulty. The settings of parameters are underspecified or subjectively determined; the training/evaluation data are not widely available.

**Reviewer Confidence:**

3: Pretty sure, but there's a chance I missed something. Although I have a good feel for this area in general, I did not carefully check the paper's details, e.g., the math, experimental design, or novelty.

**Typos Grammar Style And Presentation Improvements:**

Note: questions given here aren't to be answered, but should provide insight for the authors as to what information/details are missing from their paper.

The proposed method is complex, many its broad explanations are short. These could be improved.

As stated in "reasons to reject", the paper severely lacks detailed examples of data and explanations of data flow, and an overall detailed explanation of the proposed method that relies on these examples.

l3 "the model": no model was introduced yet

l211-213: unclear (surely related to lack of examples)

l229-233: so, since knowledge is given, this section talks about the training of the teacher models, right ? This is unclear.

Are un-instanciated arguments given as-is ? If so, are these words trainable ?

l292-293 "the teacher": which one ?

l305-306: this entails that there are several steps to training, which wasn't said until then.

l319-321: sentence without a verb

l429-441: are these variations of the proposed methods ? If so, this is unclear, and I'm also unsure which one has which component of the proposed method.

---

> ### Author Rebuttal · Authors · 2023-08-25
>
> Thank you very much. We feel sorry for some misunderstandings due to the lack of details, please give us a chance to explain them one by one.
>
> **Q**: How do you obtain the templates?   \
> **A**: The raw template is manually designed by inserting special pseudo tokens, where the slots remain empty. The knowledge-enriched template is automatically constructed by filling the raw template’s slots with ground-truth event type or event arguments.
>
> **Q**: Is the template used by the student model too?\
> **A**: The student and teacher use raw template and knowledge-enriched template, respectively. The raw template does not contain any ground-truth information, while the knowledge-enriched template contains some ground-truth information. (We adopt knowledge distillation to transfer knowledge from teacher to student, which can drive a well-formed student model to perform causality reasoning and alleviate the unavailability of ground-truth information in the test stage.)
>
> **Q**: How to justify your method with template while other baselines do not?\
> **A**: The generation-based baselines also utilize template as part of the input, which is consistent with our method. Therefore, it is fair to compare baselines with our method.
>
> **Q**: What are details of the rule-matching algorithm?\
> **A**: The rule-matching algorithm uses regular expressions to match the text span between special pseudo token pair from the output sequence as corresponding event type or argument, e.g., we can get “Price Increase” as cause event type from “<Cause><type>Price Increase</type>...”.
>
> **Q**: Is the "context" the same as the "text"?\
> **A**: Yes, “context” and “text” have same meanings here.
>
> **Q**: Does l229-233 talk about the training of the teacher models?\
> **A**: This is the ground-truth output for student and teacher during the training stage, which is used to compute the generation loss. The ground-truth output is constructed by filling **all** the slots in the raw template with ground-truth labels.
> While the construction of the template is in the input stage. The knowledge-enriched template is constructed by filling **certain** slots according to the teacher's functions (The event argument extractor teacher needs to fill ground-truth event type of cause-effect event; The effect event extractor teacher needs to fill ground-truth event type and arguments of cause event.)
>
> **Q**: Are un-instantiated arguments given as-is?\
> **A**: The un-instantiated arguments in raw template remain empty in the input stage and the model is trained to fill the slots with ground-truth labels.
>
> **Q**: Which one does "the teacher" in l292-293 refer to?\
> **A**: Either. For convenience, we use the same equation to describe the distillation process of the two teachers, because they only have different input templates, and the distillation method is the same.
>
> **Q**: What are the training steps of the method?\
> **A**: We first train teacher with knowledge-enriched template. Then we train student model with raw template and the supervision of the teacher. The parameters of the teacher model are fixed when training student.
>
>
> **Q**: Are the generation-based methods variations of the proposed method?\
> **A**: BART-ECE, CE-Pipeline, and TA-Pipeline are additionally proposed generation-based methods for comparison either in a joint or pipelined way. Student and MKD are variant methods, where Student is the generative model without the supervision of teachers, and MKD adopts raw template to train teachers instead of knowledge-enriched template. We will give a more detailed descriptions in Appendix.

---

### Meta-Review · Area_Chair_UJSi · 2023-09-20

**Recommendation:** 5

**Metareview:**

This paper formulates event causality extraction as a template-based conditional generation problem to capture interactions between the cause and effect event. All the reviewers agree that this paper provides the sound method and makes significant contributions. The authors also give the detailed responses to the questions. The reviewers suggest some valuable methods for improving the paper. It would be nice if the authors could take them into consideration.

---

### Decision · Program_Chairs · 2023-10-07

**Decision:**

Accept-Main

**Comment:**

This paper formulates event causality extraction as a template-based conditional generation problem to capture interactions between the cause and effect event. All the reviewers agree that this paper provides the sound method and makes significant contributions. The authors also give the detailed responses to the questions. The reviewers suggest some valuable methods for improving the paper. It would be nice if the authors could take them into consideration.